# Effect of Substrate-RF on Sub-200 nm Al_0.7_Sc_0.3_N Thin Films

**DOI:** 10.3390/mi13060877

**Published:** 2022-05-31

**Authors:** Michele Pirro, Xuanyi Zhao, Bernard Herrera, Pietro Simeoni, Matteo Rinaldi

**Affiliations:** Electrical and Computer Engineering Department, Northeastern University, Boston, MA 02115, USA; zhao.xuan@northeastern.edu (X.Z.); herrerasoukup.b@northeastern.edu (B.H.); p.simeoni@northeastern.edu (P.S.); m.rinaldi@northeastern.edu (M.R.)

**Keywords:** scandium-doped aluminum nitride, ferroelectric, MEMS, substrate-RF, residual stress, coercive field, leakage current

## Abstract

Sc-doped aluminum nitride is emerging as a new piezoelectric material which can substitute undoped aluminum nitride (AlN) in radio-frequency MEMS applications, thanks to its demonstrated enhancement of the piezoelectric coefficients. Furthermore, the recent demonstration of the ferroelectric-switching capability of the material gives AlScN the possibility to integrate memory functionalities in RF components. However, its high-coercive field and high-leakage currents are limiting its applicability. Residual stress, growth on different substrates, and testing-temperature have already been demonstrated as possible knobs to flatten the energy barrier needed for switching, but no investigation has been reported yet on the whole impact on the dielectric and ferroelectric dynamic behavior of a single process parameter. In this context, we analyze the complete spectrum of variations induced by the applied substrate-RF, from deposition characteristics to dielectric and ferroelectric properties, proving its effect on all of the material attributes. In particular, we demonstrate the possibility of engineering the AlScN lattice cell to properly modify leakage, breakdown, and coercive fields, as well as polarization charge, without altering the crystallinity level, making substrate-RF an effective and efficient fabrication knob to ease the limitations the material is facing.

## 1. Introduction

After Akiyama [1] demonstrated the large enhancement of the piezoelectric coefficient by doping AlN with scandium, a growing number of studies have been conducted to exploit AlScN in MEMS, with particular interest in RF applications [2,3,4,5,6,7,8]. The augmented d_33_, along with a reduction in the stiffness, enables AlScN-based resonators with a higher electro–mechanical coupling coefficient. This translates to filters with larger bandwidths as compared with their AlN counterparts, while maintaining a similar fabrication flow [9]. After a few years, in 2018, Fichtner [10] demonstrated a robust and repeatable ferroelectric behavior within the material, which opened new paths to multi-functional MEMS by combining the improved piezoelectric performance with the memory capability within the same BEOL-compatible process flow. Examples are the switchable FBARs [11,12], diodes [13], and ferroelectric transducers [14]. Nonetheless, the ferroelectric properties still face large limitations in their applicability. Despite the large polarization values, if compared with other ferroelectric materials [15], the coercive field and leakage current are still too high for practical integration. While a few works addressed the latter issue [16,17], several groups demonstrated how it is possible to reduce the coercive field by acting on either the AlScN structure or on the experimental set-up. A higher Sc-content [10], higher crystallinity [18], the use of different substrates [19], and testing-temperature [20] have been proven to reduce the voltage needed to switch, while still maintaining large polarization values. Another effective tuning parameter is represented by the residual bulk stress on the film, which can be controlled though different process parameters, such as pressure, N_2_/Ar ratio, or substrate bias [21,22,23]. In 2013, while analyzing Ga_(1−x)_Sc_x_N, Zhang [24] suggested how substrate-induced mechanical stress can have a similar effect as the Sc-doping, both indeed leading to an increase of the internal parameter u, which reduced the coercive field. Despite bulk stress having been extensively studied in non-wurzite ferroelectrics, showing an induced variation in both the dielectric and ferroelectric behavior [25,26,27,28], only a few works show measured results from AlScN. In particular, ref. [10,21,29] demonstrate the linear dependence between coercive field and bulk stress, confirming Zhang’s theory, but no general insights on the overall impact of the stress have been reported. In this context, we started from different levels of the substrate-RF applied during the deposition of sub-200 nm Al_0.7_Sc_0.3_N within a co-planar sputtering module to compare the output films in a range of parameters from stress level to dielectric and ferroelectric properties in order to map the complete effect on the material from a single process parameter. After describing the fabrication flow and characterization method, the paper will focus on the extraction of the residual stress, breakdown fields, leakage currents, coercive fields, and polarization dynamics, demonstrating the large impact of substrate-RF on the overall AlScN behavior. We will show how substrate-RF induces stress levels on the films ranging from 500 MPa to −2 GPa, resulting in an increase of the c-axis dimensions of the AlScN cell, which is reported to degrade the breakdown–coercive field ratio and losses. Next, we demonstrate how the polarity of the films increases the impact on the static leakage current when smaller c-axis values are present, enabling different levels of current emission according to the state of the films. Last, the Nucleation-Limited-Switching (NLS) model [30,31] will be exploited to compare the ferroelectric kinetics, demonstrating the same switching mechanism among the films, with larger polarization reported for larger c-axis values.

## 2. Materials and Methods

### 2.1. Fabrication Flow

Several Al_0.7_Sc_0.3_N (AlScN) thin films were deposited from a 12” alloy target installed on an Evatec Clusterline-200 PVD module onto 200 mm-Si <100> 20 Ω/sq wafers coated with 20 nm titanium, with 80 nm platinum acting as a bottom electrode layer. Before the AlScN deposition, chamber and target conditioning procedures were performed to improve the base pressure and cleanliness of the target [32,33]. The thin films were deposited starting from a base pressure of 7 × 10^−8^ mbar and a chuck temperature set at 300 °C. The wafers sat for 5 min in the hot chuck before a 6 kW pulsed DC was applied for 250 s with 90 sccm of N_2_ flow. The only variation among the films was the applied substrate-RF, which has been varied from 0 to 200 W. A final aluminum layer of 50 nm was sputtered, within the Clusterline tool, to complete the metal-ferroelectric-metal (MFM) structure. Single step photolithography and inductively coupled plasma etching were utilized to shape the top electrode features, which consisted on circular 0.144 mm^2^ pads. Lastly, access to the continuous bottom electrode was created by etching the AlScN chips in phosphoric acid heated at 150 °C.

### 2.2. Characterization Method

The in-plane residual stress was extracted with a Flexus Tencor, which measured the wafers’ bow curvature before and after the AlScN depositions. An area of 16 cm^2^ from the center of the wafer was diced to locally analyze the material through X-ray diffractometry (XRD) (gonio-scan and omega-scan) and scanning electron microscopy (SEM). The electrical performance of the fabricated MFM structures was analyzed by applying different voltage waveforms, between the top and bottom electrodes, and reading the output current through a virtual ground amplifier embedded in an Aixacct TF analyzer. First, the breakdown fields were extracted from triangular waveforms at different frequencies by observing the voltage at which the MFM structures became a short circuit. A total of 25 samples were measured for each voltage–speed combination to apply a Weibull distribution, as shown in Equation (1):(1)$$F=1-exp\left(-{\left(\frac{E}{E_{b}}\right)}^{\beta }\right)$$
where *F* is the cumulative probability of the electric failure, *E* is the experimental breakdown, *E_b_* is the breakdown field for which the cumulative probability is 63.2%, and β is the shape factor [34]. Next, the dielectric behavior was studied by applying 1 V_pp_ sine waves to pristine MFM structures to extract tanδ and dielectric permittivity (ε) in the kHz range. Afterwards, the DC response was evaluated though current density measurements, which allowed us to isolate the resistive behavior within the dielectric. A 50 V bi-polar, bi-directional voltage staircase with a 1 V step and 2 s holding time was applied to the samples at 25 °C, 35 °C, and 45 °C. The current response at each voltage step was extracted by averaging the output from 70% to 90% of the step length (2 s) to filter out any reactive behavior. The analysis was applied to pristine capacitors, and it started from the negative bias to induce a polarity inversion within each sweep. The measured data were then plotted as ln(J/T^2^) vs. √E and ln(J/E) vs. √E in order to discriminate the limiting leakage mechanism, i.e., the interface-limited Schottky emission [16] or bulk-limited Pool–Frenkel [13].

Next, the ferroelectric properties were analyzed, focusing on the negative-to-positive switching since the higher leakage shown in the reverse direction overcame the current monitoring limit of the instrument. Despite this, a comparison among the wafers’ response, and hence a study of the effect of the substrate-RF on the ferroelectric properties, was still possible. First, the coercive fields were studied by applying bi-polar trains of triangular pulses, with maximum voltage close to breakdown to induce a full polarization, within the films. The field values were extracted for different input voltage frequencies spanning from 10 Hz to 5 kHz. Second, the polarization extraction consisted in applying a modified PUND procedure described in [35], which allowed us to demonstrate the memory capability and the dependence on the write signal. The procedure started with a negative reset pulse (N-R-1), close to breakdown, applied to a pristine capacitor and acting as reference starting point. A second positive pulse of variable duration and intensity was subsequently applied as the write pulse (W(v,t)). After these, positive (P-R-1), negative (N-R-2), and positive (P-R-2) reset pulses were respectively applied to the MFM structure to allow two full negative-to-positive switching events. The polarization values were extracted by comparing the currents in output from the P-R-2 and P-R-1 pulses, which corresponded to a full and a variable polarization switching, respectively. The trapezoidal read voltages consisted of 50 μs rise, plateau, and fall times, with a fixed voltage intensity, which is chosen to be right above the coercive field to decrease the impact of leakage. On the other hand, the write pulse was varied in intensity (from 30 V to 60 V) and duration (from 50 μs to 1 s) to allow a complete mapping of the ferroelectric kinetics. The measured data were than fitted though the Nucleation-Limited-Switching (NLS) model, which showed better agreement than the classical Kolmogorov–Avrami–Ishibashi (KAI) approach [36,37]. In the latter model, the polarization evolution over time (i.e., ∆P(t)/2Ps) follows Equation (2):(2)$$\frac{\Delta P\left(t\right)}{2P_{S}}=\left[1-\exp{\left(-t/\tau \right)}^{n}\right]$$
where P_S_, τ, and n are the spontaneous polarization, switching time, and dimensionality factor, respectively. On the other hand, the NLS model assumes a Lorentzian distribution over the switching time, resulting in Equation (3):(3)$$\frac{\Delta P\left(t\right)}{2P_{S}}=\int^{\;}\left[1-\exp{\left(-t/\tau \right)}^{n}\right]F\left(\log\tau \right)d\left(\log\tau \right)$$
where:(4)$$F\left(\log\tau \right)=\frac{A}{\pi }\left[\frac{\omega }{{\left(\log\tau -{\log\tau }_{1}\right)}^{2}+\omega^{2}}\right]$$
where A, ω, and logτ are a normalized constant, half-width at half-maximum of the distribution, and median logarithmic value of the distribution, respectively [35]. 

## 3. Results

### 3.1. Material

The different substrate-RF levels induced a linear increase of the negative DC bias measured at the substrate during deposition, going from −45 V to −76 V, as reported in Figure 1a. Larger DC biases increased the bow displacement of the samples (Figure 1b), resulting in different stress levels ranging from 500 MPa to −2.2 GPa, as shown in Figure 1c.

The XRD gonio scan (Figure 2a) showed a shift in the peak corresponding to the <002> AlScN plane, suggesting a shear strain in the c-axis of the structure. No peaks corresponding to the a-plane were detected, indicating a preferred out-of-plane orientation. The full width half maximum (FWHM) of the rocking curve (RC) was 2.4° for all samples, except for RF-200, which showed RC FWHM = 3.2°, confirming the limited impact of substrate-RF on the crystallinity level of the films. SEM images of the stacks’ cross-sections are shown in Figure 2b, indicating a thickness of 190 nm with no noticeable variations among different substrate-RFs.

### 3.2. Dielectric Properties

The breakdown fields at 5 kHz, along with the extracted parameters from the Weibull fit, are shown in Figure 3a. Comparable trends over frequency are noticed among the five samples for both average field and shape factor (Figure 2b,c). The highest breakdown is found for the RF-0 sample, which shows up to 7.5 MV/cm with a 5 kHz input frequency. A high substrate-RF, and hence a higher compressive stress, reduces the breakdown field but also increases the extracted shape factor, which indicates less scattered data values.

The summary of the electrical performance in the kHz range is presented in Figure 4a,b, which show pristine tanδ and dielectric permittivity, respectively. Low substrate-RFs are characterized by lower losses (tanδ < 0.008) and ε going from 17.2 to 18.5. On the other hand, tanδ > 0.01 and ε = 16.5 are extracted for the sample RF-150 and RF-200, indicating a larger impact on the electrical response from a GPa level of stress.

Figure 4c shows the logarithmic behavior of the non-switching leakage current density among the samples. RF-0, RF-50, and RF-100 are characterized by a large asymmetry, with higher losses when a negative bias is applied. Opposite trends are noticed according to the applied bias polarity, e.g., tensile–stress states show lower losses when a positive bias is applied and larger losses with a negative bias. In order to understand the source of the leakage, the current densities are plotted with different axis to discriminate a Schottky from Poole–Frenkel emission, as described in [38]. Despite that a linear trend is found in both ln(J/T^2^) vs. √E and ln(J/E) vs. √E, Schottky emissions result in permittivities in line with the published work in [16,17], differently than the Poole–Frenkel model, which results in unrealistic values (>30). A summary of the extracted optical permittivities is reported in the Appendix A. Examples of the Schottky emission plot at 25 °C for RF-0, RF-100, and RF-200 are reported in Figure 5, while Appendix A shows the complete overview of the measured leakage for all samples at different temperatures. Each plot is composed of four curves, which come from the bi-polar, bi-directional sweeps, and they are labeled according to the combination of bias–polarity and polarization–state. The positive-down (blue) and the negative-up (yellow) curves show the leakage from sweeps which induce switching, while the remaining curves, positive-up (orange) and the negative-down (purple), derive from the bias-voltages with the same polarity as the films. 

It is clear how the gap between the blue and orange curve (positive bias), as well as the gap between the purple and yellow curve (negative bias) highly changes among the samples, indicating two distinct static resistive behaviors per each polarization, which gradually converge with increased substrate-RF to have a complete un-hysteretic leakage with RF-150 and RF-200.

### 3.3. Ferroelectric Properties

First, the ferroelectric properties were studied from high-voltage, uni-polar triangular pulses applied at different frequencies. As shown in Figure 6a, two distinct current responses arose from the first and second pulses, i.e., the switching and non-switching pulses. A peak, in correspondence with the maximum voltage and indicating a large resistive in-phase current, is noticed for both pulses, while only the switching pulse induced a second current peak, which is distinguishable among all the tested frequencies. 

The coercive fields were extracted from the ferroelectric current peak and plotted in a logarithmic scale in Figure 6b. Except for RF-150, whose fields were 1 MV/cm higher, comparable magnitudes are noticed among the first three wafers, with the RF-0 sample showing a switching field as low as 1.9 MV/cm at 10 Hz and up to 3.1 MV/cm at 5 kHz. No switching peaks were detected for the RF-200 sample. The corresponding P-E loops are presented in Figure 6c, which show a decrease in both coercive field and polarization with lower RFs. The study further analyzes the ferroelectric polarization of the films by measuring the electrical dynamics of the switching mechanism. After having identified the minimum voltage needed to fully reverse the films, the polarization charge is extracted from the current response in the output of a fixed-voltage reset pulse, P-R-1, which is applied 1 s after a variable positive write pulse (W(v,t)). Figure 7 shows the adopted train of pulses along with the measured output currents for the different samples. It is clear how the write pulse width, which varies from 50 μs to 1 s, with v = 60, (W(60,t)), has a clear impact on the polarization state of the films.

Thanks to the trapezoidal input waveform, a first analysis is possible by comparing the current peaks, which are the sum of the capacitive, resistive, and ferroelectric components, and the plateau currents, which instead filter out any time variant behavior. Similar trends are noticed among the four samples, with an increase in plateau and peak current with higher substrate-RFs, as shown in Appendix A. By varying the write amplitude along its time-width, it is possible to map the complete polarization reversal though ferroelectric switching models. As shown in Appendix A, an NLS model better described the polarization dynamic when compared with the KAI model, indicating the need to express the ferroelectric activation time through a Lorentzian distribution instead of a Delta Dirac. A summary of the extracted NLS parameters is shown in Figure 8a–c, which confirms a higher polarization for higher substrate-RFs (Figure 8a). The mean activation time and ω show similar behavior among the samples, with a linear dependence with 1/E and 1/E^2^, respectively, (Figure 8b,c), indicating a similar behavior to HZO [35] and poly-PZT [39].

## 4. Discussion

The substrate-RF has been demonstrated to have a direct impact on the films’ structure and properties. Higher RF values increase the impinging energy of the specimen at the substrate [22], resulting in larger compressive states and a larger out-of-plane lattice dimension (Figure 9a). Even if more investigations are needed to map the whole effect on the lattice structure, i.e., the a-axis, the reported variations in the dielectric and ferroelectric domains evidence the critical role of a single process parameter in the deposition of AlScN thin films. Overall, low substrate-RFs, and hence tensile states, showed better performance, with a higher breakdown coercive field ratio and lower losses while maintaining a polarization higher than 130 μC/cm^2^, as summarized in Figure 9b,c. 

Furthermore, the induced positive stress is reported to increase the gaps between the polarization-dependent current emissions, resulting in larger hysteresis on the ln(J/T^2^) vs. √E capacitor characteristic. As described in [13,16,17], the AlScN polarity highly affects the band alignment with the metals, which results in different barrier heights and in a clear shift in the leakage current for both positive and negative bias. Differently than [16], which was based on a symmetric capacitor structure, we report a preferred orientation state for the AlScN films, i.e., they are characterized by lower losses, which corresponds to the as-deposited polarity, unchanged among all the samples. A zoom-in of the different resistive behavior per different polarization states is presented in Figure 10a, which helps highlight the horizontal jumps corresponding to the negative-to-positive switching. As indicated by the arrows in Figure 10a, the measured shift is reported to decrease with the increase of substrate-RF until having a completely un-hysteretic leakage for RF-150 and RF-200 samples, which suggests a weaker and weaker impact of the MFM state on the contact interface. While the slopes remains invariant, i.e., with the same optical permittivities per different states, the curve intercepts, (=ln(A × T_t_) + qΦ_Bn,app_/(K_b_ T)), change according to the MFM state, indicating a variation in the apparent Schottky barrier height (Φ_Bn,app_) and in the effective Richardson constant (A × T_t_). Such variations are quantifiable by plotting the extracted curve intercepts per different temperatures, as shown in Figure 10b and Appendix A for the positive-up and positive-down combinations, respectively [16,17]. Figure 10c summarizes how the lower the substrate-RF, the larger the impact of the polarization state on both Φ_Bn,app_ and A × T_t_., i.e., on the static behavior of the MFM structure. Compatible findings are obtained within the ferroelectric characterization in both coercive field and polarization extractions. As in the leakage analysis, on which low substrate-RFs induced lower fields to transition from the two resistive states, the PUND-based measurements demonstrated lower coercive fields with tensile stresses, confirming the previous works on the effect of stress on the switching field [10,29]. 

Additionally, the switching currents, in output of the same input voltage, exhibit not only different voltage thresholds (i.e., voltage for which the current rises exponentially) but also different levels of saturation current (i.e., current level at the end of the trapezoidal plateau) per different polarization states, confirming the varying resistive behavior within the MFM structure. As in the leakage, such variations tend to decrease with higher substrate-RFs until they have similar voltage thresholds and plateau currents per different polarization state. It is worth mentioning how the RF-150 sample, despite an un-hysteretic leakage behavior, still shows a switching current dependent on the write signal, differently than RF-200, which does not show any sign of ferroelectric switching. The extracted NSL parameters demonstrate how the induced deformations not only vary the resistive behavior within the capacitors, but the whole switching dynamics as well. Like HZO [35] and poly- PZT [39], a linear relationship between logτ with 1/E and ω vs. 1/E^2^ is noticed in the AlScN films, indicating a similar predominant switching mechanism among the materials. In particular, compressive states are reported to increase both the slopes of the logτ vs. 1/E and ω vs. 1/E^2^ characteristics, suggesting slower and broader switching mechanism for intermediate polarization states, which is a sign of an increase in the pinning sites within the film. Future works will focus on the impact of electrode size, as well as on the investigation of the RF-150 sample, which does not follow the trend and shows a very small value of omega, which was constant over the different applied fields.

## 5. Conclusions

The paper demonstrates the effect of substrate-RF on 200 nm thin films deposited from a 12” Al_0.7_Sc_0.3_ alloy target on a platinum substrate. From macroscopic to microscopic properties, this work compares five films deposited with different RF levels, which resulted in an induced DC bias at the substrate from −40 V to −76 V. Such a variation of energy leads to an increase of the negative bow of the wafers, which corresponded to higher compressive stresses of the films. This leads to different c-axis dimensions of the AlScN lattice structure. An overall decrease of dielectric and ferroelectric properties with an increasing c-axis was noticed. In particular, even if an undistorted lattice (and hence low stress) is preferred within the released MEMS devices to avoid cracks and peeling, we demonstrate substrate-RF as valuable knob to decrease losses and increase the breakdown to coercive field ratio, while still maintaining high polarization values. Furthermore, low substrate-RF has been demonstrated to enhance the effect of the MFM state on the contact barrier, producing two distinct resistance paths per each polarization direction. In conclusion, we demonstrate the high dependence of both dielectric and ferroelectric properties on substrate-RF, describing its potential in obtaining integrable ferroelectric AlScN thin films.

## Figures and Tables

**Figure 1 micromachines-13-00877-f001:**
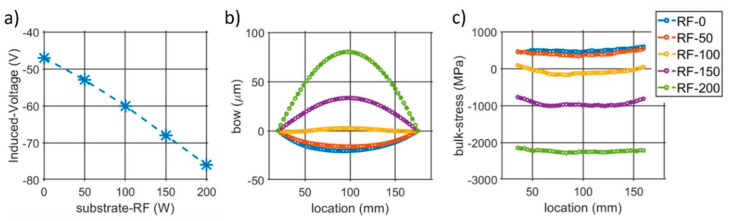
(**a**) Induced substrate DC biases per different RFs. (**b**) Wafers’ curvatures measured with a Flexus Tencor after AlScN depositions and (**c**) corresponding bulk stresses extracted though Stoney equations.

**Figure 2 micromachines-13-00877-f002:**
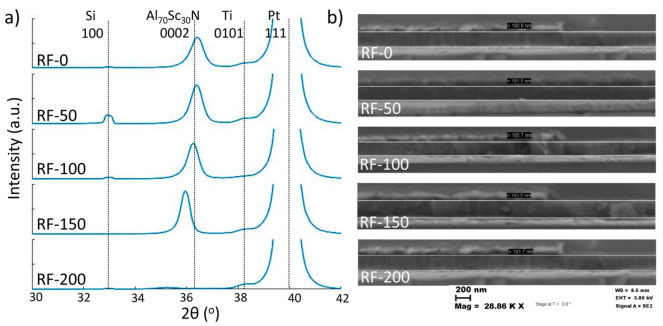
(**a**) Gonio-scan of the five samples showing the induced shifts of the 0002 AlScN plane. The dotted line represents the theoretical reference [18]. (**b**) SEM cross-sections of the samples with an average thickness of 190 nm.

**Figure 3 micromachines-13-00877-f003:**
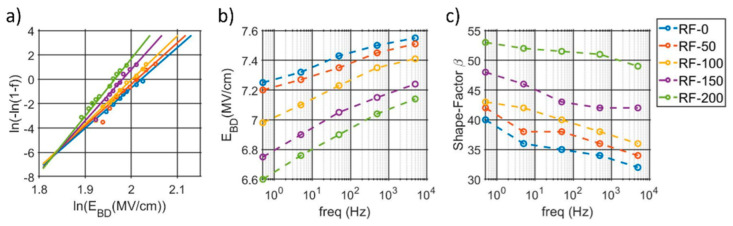
(**a**) Weibull distributions of the breakdown fields with a 5 kHz input voltage for the five samples. (**b**) Average breakdown fields and (**c**) shape factors extracted at different frequencies.

**Figure 4 micromachines-13-00877-f004:**
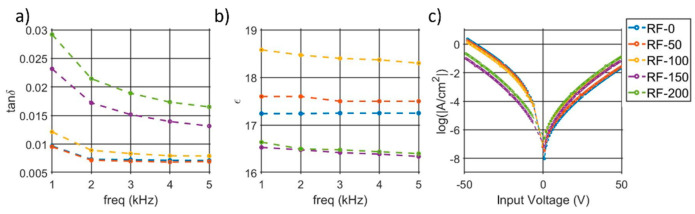
(**a**) Dielectric losses and (**b**) dielectric permittivity measured from 1 to 5 kHz. (**c**) Leakage current density of the non-switching sweep.

**Figure 5 micromachines-13-00877-f005:**
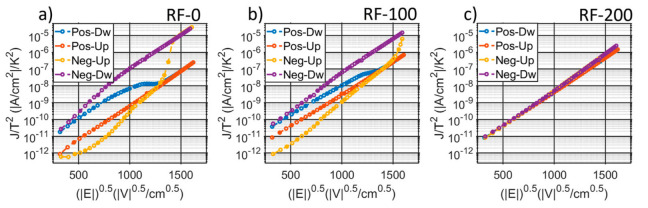
Example of leakage current densities for (**a**) RF-0, (**b**) RF-100, and (**c**) RF-200 measured at 25 °C and plotted versus the square root of the field to indicate the linear relationship between ln(J/T^2^) and √E, typical of Schottky emissions. Each leakage measurement considers the combination of bias (positive or negative) and film polarity (up or down). Appendix A in the Appendix A summarizes the measured leakage for all the sample at three different temperatures.

**Figure 6 micromachines-13-00877-f006:**
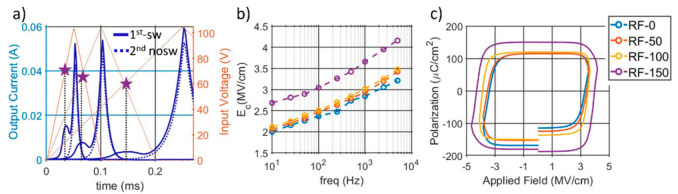
(**a**) Examples of switching (continuous) and non-switching (dotted) currents (blue curves) for different input frequencies, along with the corresponding coercive field extraction method. (**b**) Summary of coercive fields for all the switchable samples from 10 Hz to 5 kHz. (**c**) Extracted P-E loops with a 5 kHz input voltage.

**Figure 7 micromachines-13-00877-f007:**
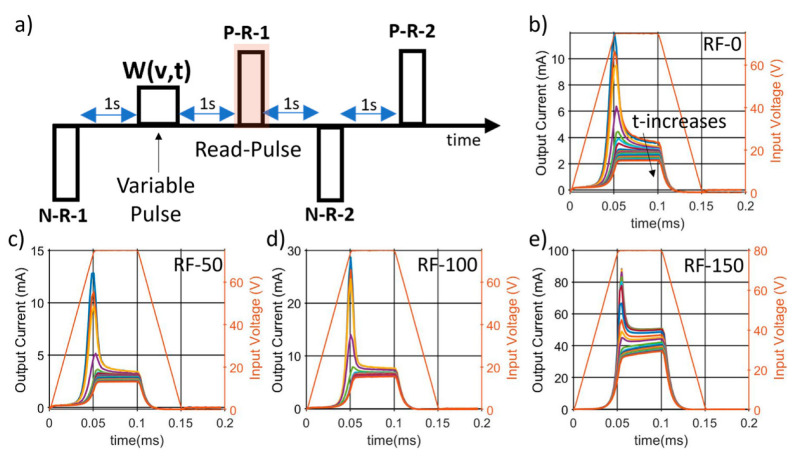
(**a**) Modified PUND method to analyze the polarization dynamic of the films. (**b**–**e**) Corresponding output read currents for different time-widths of the write pulse, from 50 μs to 1 s, for all the switchable samples: (**b**) RF-0, (**c**) RF-50, (**d**) RF-100, and (**e**) RF-150.

**Figure 8 micromachines-13-00877-f008:**
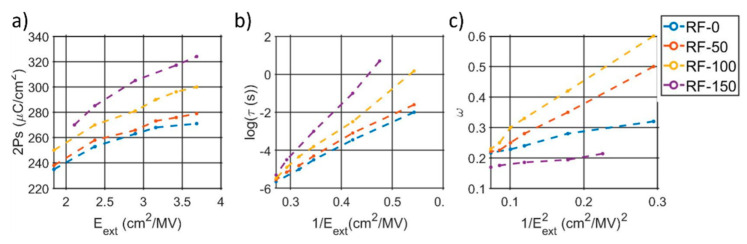
Summary of the extracted NSL model parameters: (**a**) switching polarization per different applied electric fields during the write signal, (**b**) average activation time versus the inverse of the applied field, and (**c**) relation between ω and the inverse of the square of the field.

**Figure 9 micromachines-13-00877-f009:**
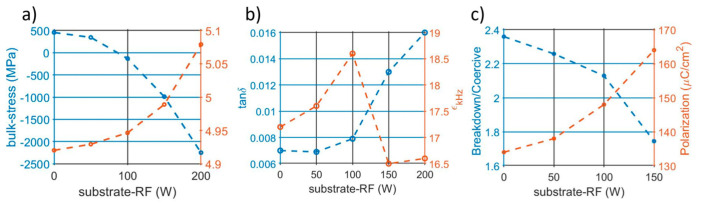
(**a**) Substrate-RF-induced variation of the residual stress extracted from the Flexus measurement and on the c-axis dimension extracted from the XRD scan [40]. (**b**) 5 kHz tanδ and dielectric permittivity of the pristine capacitors. (**c**) Breakdown/coercive field ratio and polarization variation per different substrate-RF.

**Figure 10 micromachines-13-00877-f010:**
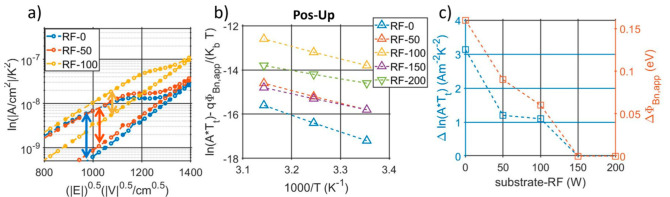
(**a**) Zoom-in of the ln(J/T^2^) vs. √E characteristic around the transition fields with positive bias applied per RF-0, RF-50, and RF-100. The arrows indicate the difference between the two resistive states of the films corresponding to the two polarity states of the films. (**b**) Extracted intercepts per different temperatures to evaluate the induced barrier variations with a positive bias applied to the up-polarization state. Appendix A shows the intercept variations corresponding to the opposite polarity state. (**c**) Impact of the substrate-RF on the polarization-dependent resistive behavior: the left axis shows the difference between the extracted up-polarity ln(A × T) and down-polarity ln(A × T), while the right axis plots the same difference calculated for the apparent barrier.

## Data Availability

Data are available within the article.

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
