# Peer review of "Effect of Substrate-RF on Sub-200 nm Al0.7Sc0.3N Thin Films"

_micromachines, 2022, doi:10.3390/mi13060877_

Round 1

Reviewer 1 Report

AlScN films, 200 nm thick, were sputtered onto Pt substrates by sputtering technique; the effects of one growing process parameter, the substrate RF, onto the films properties (from stress-level to dielectric and ferroelectric properties) were investigated.  The obtained results demonstrate the large impact of the substrate RF  onto residual stress, breakdown fields, leakage currents, coercive fields, and polarization dynamics of the AlScN layers.

the paper is well organized and clear; as to my opinion, it can be published as it is.

Author Response

Thank you for reviewing the articles

Reviewer 2 Report

In this paper, the author research the effect of substrate-RF on sub-200nm al0.7sc0.3n thin films, and give corresponding characteristics. The dielectric and ferroelectric properties are discussed in details with sufficient analysis. Here are my comments:

1.     Format and typo: the equations in this paper are not unified in the same font; the figures are plotted in different size and resolution, Figure 5 is too small compared with Figure 6; line 213 and line 214, the ‘gap’ should be gaps; in Figure 7, there are two Figure 7-C;

2.     The Pt-ref work as a comparison in Fig 1-b, while in fig. 1-c, the Pt-ref is missing. Are there any persuasive enough proof to remove the Pt-ref?

3.     In Fig2-a, the XRD pattern seems to has peaks go outside the figure. And the XRD pattern should be dealing with the noise deduction and plot separately since the XRD peak shift is not clear.

4.     In line 184, the author refers the highest breakdown shows up to 7.5 MV/cm with 5K Hz input frequency. However, in Figure 3 b, the 7.5 MV/cm corresponded with 10^3 log- frequency, which is not 5K. it should be explained the reason for this mismatch.

5.     The ferroelectric hysteresis loops should be measured to discuss the ferroelectric properties.

6.     In line 303, the author refer that the temperature characterizations are needed. Due to the importance discussed by the authors, it should contain sufficient temperature characterizations in this paper.

Round 2

Reviewer 2 Report

After revision, the authors well addressed my concerns. So I recommend the paper can be accepted.